# Matere Bonds vs. Multivalent Halogen and Chalcogen Bonds: Three Case Studies

**DOI:** 10.3390/molecules27196597

**Published:** 2022-10-05

**Authors:** Rosa M. Gomila, Antonio Frontera

**Affiliations:** Department of Chemistry, Universitat de les Illes Balears, Crta. de Valldemossa km 7.5, 07122 Palma de Mallorca, Baleares, Spain

**Keywords:** matere bond, chalcogen bond, halogen bond, X-ray structures, DFT calculations

## Abstract

The term matere bond has been recently used to refer to an attractive noncovalent interaction between any element of group 7 acting as an electrophile and any atom (or group of atoms) acting as a nucleophile. The utilization of metals such as σ-hole donors is starting to attract the attention of the scientific community. In this manuscript, a comparison between matere bonds and well-known σ-hole interactions (halogen and chalcogen bonds) is carried out using three X-ray structures, retrieved from the Cambridge structural database (CSD), and density functional theory calculations (DFT). The novelty of this work resides in the utilization of a neutral Re(VII) system as the matere bond donor and multivalent chalcogen and halogen donors. In fact, as far as our knowledge extends, the description of σ-hole interactions in Se(VI) is unprecedented in the literature. The σ-hole interactions in Re(VII), Se(VI) and Cl(VII) electron acceptors are analyzed and compared using several computational tools.

## 1. Introduction

The study and utilization of noncovalent interactions involving metal centers are attracting the attention of the scientific community. In particular, theoreticians and crystallographers are putting effort into understanding the physical nature of such contacts [1,2,3,4,5,6,7,8,9,10]. Starting from the pioneering works of Brinck et al. on regium bonds (group-11) [1,2], the field has been extended toward group 12 (spodium bonds) [3,4], group 6 (wolfium bonds) [5], group 7 (matere bonds) [6] and group 8 (osme bonds) [7,8,9]. Apart from the relevance of these interactions in the solid state and in biological systems [10], it has been demonstrated that regium bonds are also important in catalysis [2].

The extension of the σ-hole concept [11,12] from the p-block to the d-block of the periodic table, along with the necessity of using a different name for each group, is not understood by part of the scientific community. This is especially true for the organometallic and coordination chemistry community. In fact, most of the literature dealing with non-covalent interactions involving metals describes the interaction as coordination or semico-ordination bond. On the other hand, it has been proposed that noncovalent interactions, wherein it is possible to identify an element playing the role of an electrophile, are termed by using the name of the group of the periodic table the electrophilic atom belongs to [13]. This terminology generalizes the International Union of Pure and Applied Chemistry (IUPAC) criterion used for the definition of hydrogen [14], halogen [15] and chalcogen bonds [16]. Systematic and unambiguous periodic naming is important in any discipline of science.

In this manuscript, a comparison of matere bonds (MaB) to multivalent chalcogen (ChB) and halogen bonds (HaB) is reported to emphasize the similitudes between the matere bonds and the well-established and recognized halogen and chalcogen bonds (see Figure 1) and also to demonstrate that the term “(semi)coordination bond” is not always adequate to define donor–acceptor interactions involving metals. All three structures shown in Figure 1 form centrosymmetric self-assembled dimers in the solid state where two symmetrically equivalent Y···O interactions are established (Y = Re, Se, Cl). Several computational tools have been used to analyze the interactions, including molecular electrostatic potential (MEP) surfaces, the noncovalent interaction plot (NCIplot) and natural bond orbital (NBO) analysis. These X-ray structures were selected because they form very similar assemblies in the solid state; therefore, they are ideal for comparing halogen and chalcogen bonds (p-block σ-holes) to matere bonds (d-block).

## 2. Results and Discussion

### 2.1. Description of the X-ray Structures

The three X-ray structures studied in this work are shown in Figure 2. The Re(VII) compound selected for this study is HETRUT, which was reported in 1999 by Gosink et al. [18] and synthesized by reacting O(t-Bu_2_SiOH)_2_ with Re_2_O_7_. The molecular structure of HETRUT (Figure 2a) reveals two ReO_4_ units, each attached to one silicon atom. The environment of rhenium is tetrahedral. In the solid state, this compound forms 1D supramolecular polymers that propagate via the formation of Re···O contacts. Two different binding modes are observed: on one end, the O-atom of the adjacent molecule is located opposite the single O–Re bond (3.602 Å), and on the other end, the O-atom is located opposite the O=Re double bond (3.324 Å, see Figure 2a). The latter is more directional (171.8°) than the former. In the binding mode where the interaction is opposite the O–Re single bond, the distance is almost identical to the sum of van der Waals radii (ΣR_vdw_), and in the other one, the Re···O distance is much shorter (3.234 Å), thus suggesting a stronger interaction. In any case, both Re···O distances are much longer than the sum of covalent radii (ΣRcov = 2.16 Å), thus confirming the noncovalent nature of the Re···O interactions (matere bonds).

The X-ray structure of WABPAR that was synthesized and X-ray characterized in 2003 by Ritchera et al. by mixing a large excess of diethyl ether with selenium trioxide at a low temperature (−116 °C) [19] is shown in Figure 2b. Pairs of Et_2_O·SeO_3_ molecules form self-assembled dimers governed by chalcogen bonding interactions (Se···O 3.112 Å), which are shorter than the sum of the van der Waals radii. To our knowledge, these types of chalcogen bonds involving Se(VI) have not been described before, though such intermolecular contacts were described by the original authors as weak interactions. The KUCRAD structure was obtained via the reaction of 1-adamantyl-1-iodomethane with an excess of lithium perchlorate [20]. In the solid state, pairs of adamantan-1-ylmethyl perchlorate molecules are held together by halogen bonds, forming self-assembled dimers. In this case, this multivalent halogen bond’s Cl···O distance is slightly longer than the ΣR_vdw_ (3.35 Å). Although multivalent halogen bonds in the +7 oxidation state such as in the KUCRAD dimer are not common, the structure-directing role of similar halogen bonds has been recently reported in periodate salts [21]. The resemblance of chalcogen and halogen bonds in the self-assembled dimers with the matere bonds in HETRUT is remarkable, and further reinforces the fact that such contacts are most likely σ-hole interactions, as further analyzed below.

### 2.2. Theoretical DFT Study

#### 2.2.1. Molecular Electrostatic Potential (MEP) Study

The MEP surfaces of the three selected compounds are shown in Figure 3. In the case of the Re(VII) derivative, three σ-holes are accessible, with those opposite the O=Re double bonds being more intense (+32.1 kcal/mol) than that those opposite the O–Re single bond (+21.3 kcal/mol). This agrees well with the interactions described above for the 1D polymer (Figure 2a), where the matere bonds opposite the double bond are shorter and more directional. The main difference in the Re compound with respect to the Se and Cl derivatives is that in the latter compounds, the σ-hole opposite the single bond instead of the double bonds is more available for establishing ChBs and HaBs. The value at the σ-hole opposite the single bond is significantly smaller for Cl(VII) (+9.5 kcal/mol) than for Se(VI) and Re(VII) (+19.8 kcal/mol and +21.3 kcal/mol, respectively), which is in line with the small ability of chlorine to participate in conventional halogen bonding [22]. The MEP minima are located in the three compounds in the O-atoms, varying from –17.6 kcal/mol in KUCRAD to –28.8 kcal/mol in WABPAR. This analysis anticipates the higher ability of Se(VI) and Re(VII) derivatives to participates in σ-hole interactions compared to Cl(VII), as further demonstrated in the following sections.

#### 2.2.2. X-ray vs. Optimized Geometries

A comparison of the fully optimized self-assembled dimers of HETRUT, WABPAR and KUCRAD with the solid-state structures (represented in Figure 2) was carried out to investigate if these dimers are stable and to know if the σ-hole distances and angles are influenced much by packing effects. Interestingly, all self-assembled dimers (see Figure 4) are stable in the gas phase, with distances that are in acceptable agreement with the experimental ones. Moreover, the experimental and theoretical O···Y–O (Y = Re, Se, Cl) angles are in good agreement. The main difference is observed in the chalcogen bond dimer (WABPAR) where the theoretical chalcogen distance is 0.23 Å shorter than the experimental one. Nevertheless, the minimum nature of these isolated dimers strongly suggests that the observation of such assemblies in the solid state is not simply due to packing effects.

#### 2.2.3. Energetic and NCIplot Analyses

In order to further characterize the matere, chalcogen and halogen bonds in the self-assembled dimers, noncovalent interaction plot (NCIplot) analysis was used. This method, based on electron density (ρ), is useful for revealing noncovalent interactions in real space. In the NCIplot, the reduced density gradient (RDG) isosurface is represented, and the sign of the middle eigenvalue of the Hessian of ρ times ρ (i.e., (signλ_2_)*ρ) is mapped onto the surface using a color scale. In this work, blue and green colors are used for strong and weak interactions. Moreover, red and yellow colors are used for strong and weak repulsive interactions, respectively. Figure 5 shows the NCIplot analysis of all the dimers, showing very similar RDG isosurfaces. Interestingly, in all cases, the RDG isosurface forms a disk between the interacting atoms, thus characterizing the σ-hole interactions. The green color of these disks suggests a weak nature of the NCIs. In the case of the self-assembled dimer of HETRUT shown in Figure 5, the NCIplot analysis discloses the existence of additional CH···O interactions, since extended and green RDG isosurfaces appear, upon dimerization, between the O-atoms of the ReO_4_ unit and the H-atoms of the methyl groups. The dimerization energies using the fully optimized dimers and the X-ray geometries are also gathered in Figure 5. It can be observed that the dimerization energies are quite modest for KUCRAD (HaB) in line with the long distances, the small MEP value at the σ-hole and the small size of the RDG disks between the O- and Cl-atoms. For the WABPAR dimer, the dimerization energy is modest for the experimental geometry (–1.9 kcal/mol) and moderately strong for the optimized one (–5.0 kcal/mol) in line with the shortening of the ChB contacts in the optimized geometry. Interestingly, the interaction energies for both the experimental and theoretical geometries are quite similar for the dimer of HETRUT shown in Figure 5a, thus strongly supporting that the matere bonds are not simply due to packing effects. For the dimer of HETRUT shown in Figure 5b, the interaction energy is significantly more negative than the rest (for both the experimental and theoretical geometries). This is due to the contributions of the ancillary H-bonds, and also the stronger nature of the matere bonds, since the σ-hole opposite the O=Re double bond is more positive than that opposite the single O–Re bond (*vide supra*). Finally, in all cases, the dimerization energies are smaller (in absolute value) for the X-ray geometries, due to the fact that for optimization, only the isolated dimer in the gas phase is considered; thus, it lacks the influence of the adjacent molecules, contrary to the situation in the solid state.

#### 2.2.4. NBO Analysis

A very characteristic feature of σ-hole bonding is the existence of an LP→σ* donor–acceptor interaction. This feature is very useful for differentiating noncovalent bonds from coordination bonds in those contacts involving metals. The most-used methodology to analyze such orbital interactions is the natural bond orbital (NBO). In particular, second-order perturbation analysis (see the theoretical methods below) allows researchers to identify the donor–acceptor orbitals and the associated stabilization energy. The analysis was performed for the strongest Re dimer (opposite the Re=O). The donor and acceptor NBOs are represented in Figure 6 along with the stabilization energy. Remarkably, the analysis evidences an LP(O)→σ*(Re–O) orbital interaction with a concomitant stabilization of E^(2)^ = 1.6 kcal/mol (for each Re···O contact). It can be observed that the filled LP orbital of the O atom points to the antibonding σ*(Re–O) orbital, as expected in a typical σ-hole interaction. The total orbital contribution (3.2 kcal/mol) is significant compared to the total interaction energy, revealing that, apart from the electrostatic force and ancillary CH···O contacts, charge transfer effects are also relevant.

## 3. Materials and Methods

Both the single-point calculations and geometric optimizations of the self-assembled dimers were carried out using the Turbomole 7.2 program [23]. The Cartesian coordinates are given in the Appendix A. The level of theory used for both types of calculations was PBE0-D3/def2-TZVP [24,25,26]. For rhenium, the def2-TZVP basis set was used that includes effective core potentials (ECP) and relativistic effects for the inner electrons [26]. This level of theory has been used before to study matere bonds [6]. The MEP surface plots were computed at the PBE0-D3/def2-TZVP level of theory and the 0.001 a.u. isosurface. The NCIplot [27] method was used to characterize the noncovalent interactions by means of the reduced density gradient (RDG) isosurfaces, which were plotted using the VMD program [28]. The settings for the RDG plots were: *s* = 0.45 a.u.; cut-off ρ = 0.05 a.u.; and color scale –0.04 a.u. ≤ sign(λ_2_)ρ ≤ 0.04 a.u. Natural bond orbital (NBO) [29] analysis was carried out using the NBO7.0 program [30], at the same level of theory.

## 4. Conclusions

The theoretical results reported herein, in combination with the X-ray structures, consistently demonstrate the attractive interaction and structural-directing role of matere bonds in the Re(VII) compound. Moreover, by using two similar X-ray crystal structures of Se(VI) and Cl(VII), it is evidenced that the matere bond is comparable to multivalent chalcogen and halogen bonds. Finally, the presence of σ-holes opposite the O=Re and O–Re bonds and the participation of the antibonding σ*(O–Re) orbital in the matere bond are evidenced using MEP surface and NBO analyses, thus strongly supporting the σ-hole nature of the interaction. Moreover, the interaction energies and geometries of the self-assembled dimers are similar in the solid state and in silico, thus indicating that these types of contact are not a consequence of crystal packing and, on the contrary, they have a structure-directing role.

## Figures and Tables

**Figure 1 molecules-27-06597-f001:**
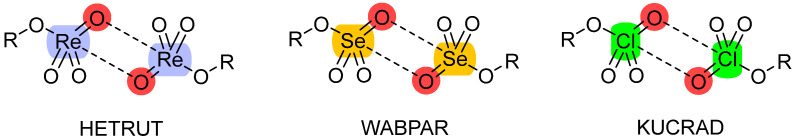
Three structures studied in this work and the CSD [17] reference codes.

**Figure 2 molecules-27-06597-f002:**
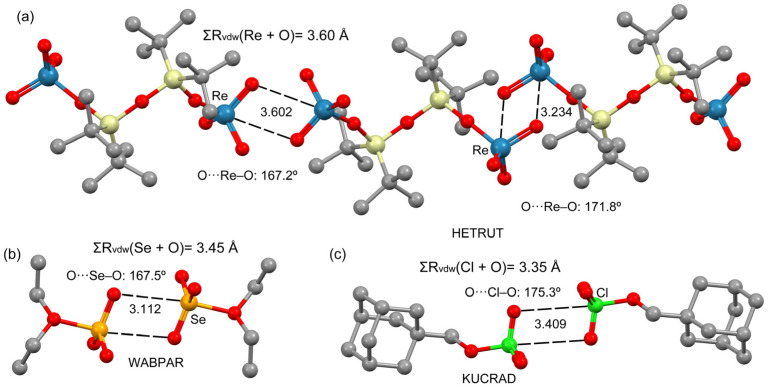
Partial views of the X-ray structures of HETRUT (**a**), WABPAR (**b**) and KUCRAD (**c**). H-atoms omitted for clarity. Distances in Å. The sums of Batsanov’s van der Waals radii (R_vdw_) are also indicated.

**Figure 3 molecules-27-06597-f003:**
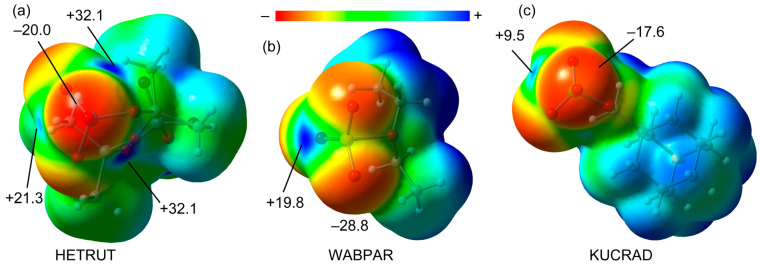
MEP surfaces of HETRUT (**a**), WABPAR (**b**) and KUCRAD (**c**). The energies at selected points of the surfaces are given in kcal/mol.

**Figure 4 molecules-27-06597-f004:**
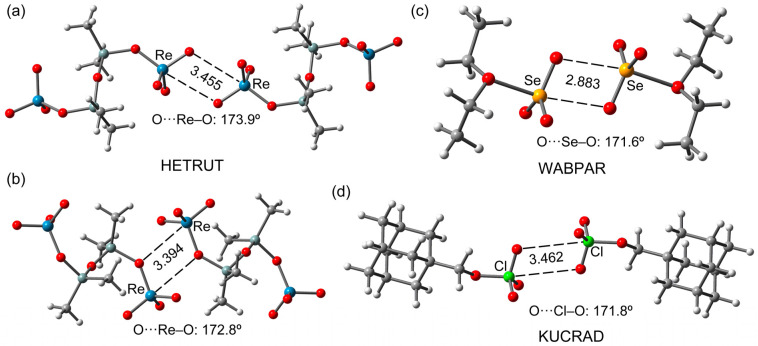
Fully optimized dimers of HETRUT (**a**,**b**), WABPAR (**c**) and KUCRAD (**d**). Distances in Å.

**Figure 5 molecules-27-06597-f005:**
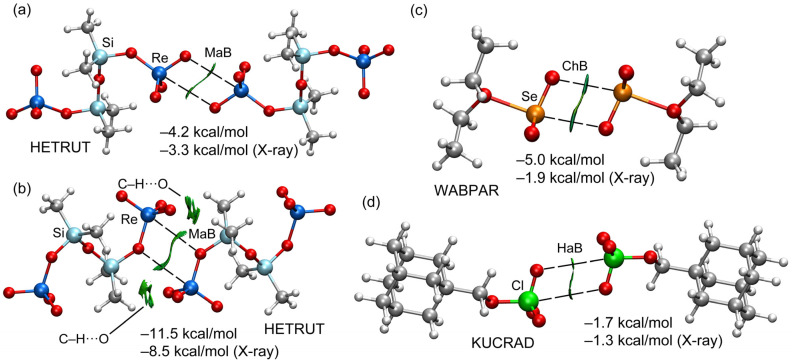
NCIPlot analyses of the dimers of HETRUT (**a**,**b**), WABPAR (**c**) and KUCRAD (**d**). The dimerization energies using both the optimized and X-ray geometries are indicated at the PBE0-D3/def2-TZVP level of theory. Isovalue for the RDG isosurface 0.45.

**Figure 6 molecules-27-06597-f006:**
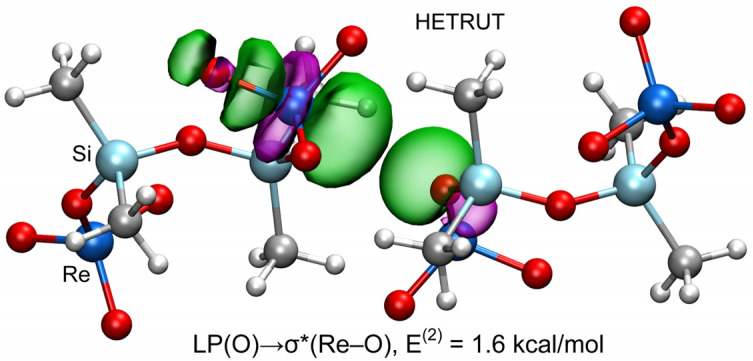
Representation of the filled and empty NBOs corresponding to the LP(O)→σ*(Re–O) donor–acceptor interaction in the MaB dimer HETRUT. The isosurface used for the MOs is 0.008 a.u.

## Data Availability

Not applicable.

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
