# Peer review of "Matere Bonds vs. Multivalent Halogen and Chalcogen Bonds: Three Case Studies"

_molecules, 2022, doi:10.3390/molecules27196597_

Round 1

Reviewer 1 Report

This study utilizes a neutral Re(VII) system to form a matere bond, which is a novel intermolecular interaction between a sigma-hole on the Re bond and a nucleophile. This bond was analyzed by geometries, energetics, AIM, NCI, and NBO, and also compared with the corresponding chalcogen and halogen bonds. The research work well done and discussed. I recommend its publication in Molecules.

Author Response

We thank the referee for his/her careful reading of the manuscript and encouraging comments

Reviewer 2 Report

This is a very clear and well-written paper concerning a topic of importance and of high interest.  I commend the authors on their presentation.  I have only one comment for the authors.

Since σ-hole interactions have only been mentioned in the literature for around 15 years, it would be appropriate to include a general reference on these.    One that includes mention of hypervalent σ-hole bonding is:  Politzer et al, PCCP, 15, 11178 (2013).  A second more recent paper can be found in PCCP, 19, 32166 (2017).

Author Response

We thank the referee for his/her careful reading of the manuscript, corrections and suggestions. The changes made are listed below:

Comment: Since σ-hole interactions have only been mentioned in the literature for around 15 years, it would be appropriate to include a general reference on these.    One that includes mention of hypervalent σ-hole bonding is:  Politzer et al, PCCP, 15, 11178 (2013).  A second more recent paper can be found in PCCP, 19, 32166 (2017).

Reply: Both references have been added.

Reviewer 3 Report

The manuscript “Matere bonds vs. multivalent halogen and chalcogen bonds: three case studies” by Rosa M. Gomila and Antonio Frontera, analyses the characteristics and strength of the interactions between two neutral subunits embedded in a crystalline framework in comparison with the DFT study. The comparison involves three types of noncovalent interactions: matere, halogen and chalcogen bonds.

I consider the work interesting; it is a step in the study of noncovalent interactions highlighting the importance of NCIs beyond hydrogen bonds in a crystal lattice.  The results are concise and well-written. The manuscript is clear.

The mistyping of rhenium oxidation state should be corrected: line 17 vs line 14, 55 et al.

Author Response

We thank the referee for his/her careful reading of the manuscript, corrections and suggestions. The changes made are listed below:

Comment: The mistyping of rhenium oxidation state should be corrected: line 17 vs line 14, 55 et al.

Reply: Corrected

Reviewer 4 Report

The manuscript describes the comparison of the nature of matere bonds relative to halogen and chalcogen bonds. The approach is appropriate and the results interesting. My only question is with regard to the choice of the three compounds studied, since it is not clear why these were chosen. Were they the only ones in the Cambridge Structural Database with this particular type of supramolecular synthon? If so, how relevant is this study to the broader community? A brief comment regarding the choice of the three crystal structures in the context of the importance of such interactions for Re-containing compounds is needed.

Author Response

We thank the referee for his/her careful reading of the manuscript, corrections and suggestions. The changes made are listed below:

Comment: The manuscript describes the comparison of the nature of matere bonds relative to halogen and chalcogen bonds. The approach is appropriate and the results interesting. My only question is with regard to the choice of the three compounds studied, since it is not clear why these were chosen. Were they the only ones in the Cambridge Structural Database with this particular type of supramolecular synthon? If so, how relevant is this study to the broader community? A brief comment regarding the choice of the three crystal structures in the context of the importance of such interactions for Re-containing compounds is needed.

Reply: In the case of halogen and chalcogen, the crystal structures are unique. In case of matere, there are additional X-ray structures in the CSD. The particular X-ray structures reported in the manuscript were selected because they form very similar motifs in the solid state, so they are ideal to compare Halogen and Chalcogen bonds (p-block) with the matere bond (d-block). A brief comment has been added in page 2, lines 55-57.